# In Silico Identification and Validation of Organic Triazole Based Ligands as Potential Inhibitory Drug Compounds of SARS-CoV-2 Main Protease

**DOI:** 10.3390/molecules26206199

**Published:** 2021-10-14

**Authors:** Vishma Pratap Sur, Madhab Kumar Sen, Katerina Komrskova

**Affiliations:** 1Laboratory of Reproductive Biology, Institute of Biotechnology of the Czech Academy of Sciences, BIOCEV—Biotechnology and Biomedicine Centre of the Academy of Sciences and Charles University, Prumyslova 595, 252 50 Vestec, Czech Republic; VishmaPratap.Sur@ibt.cas.cz; 2Department of Agroecology and Crop Production, Faculty of Agrobiology, Food and Natural Resources, Czech University of Life Sciences Prague, Kamycka 1176, 165 00 Prague, Czech Republic; senm@af.czu.cz; 3Department of Zoology, Faculty of Science, Charles University, Vinicna 7, 128 44 Prague, Czech Republic

**Keywords:** SARS-CoV-2, main protease, triazole, docking, MD simulation, drug

## Abstract

The SARS-CoV-2 virus is highly contagious to humans and has caused a pandemic of global proportions. Despite worldwide research efforts, efficient targeted therapies against the virus are still lacking. With the ready availability of the macromolecular structures of coronavirus and its known variants, the search for anti-SARS-CoV-2 therapeutics through in silico analysis has become a highly promising field of research. In this study, we investigate the inhibiting potentialities of triazole-based compounds against the SARS-CoV-2 main protease (M^pro^). The SARS-CoV-2 main protease (M^pro^) is known to play a prominent role in the processing of polyproteins that are translated from the viral RNA. Compounds were pre-screened from 171 candidates (collected from the DrugBank database). The results showed that four candidates (Bemcentinib, Bisoctrizole, PYIITM, and NIPFC) had high binding affinity values and had the potential to interrupt the main protease (M^pro^) activities of the SARS-CoV-2 virus. The pharmacokinetic parameters of these candidates were assessed and through molecular dynamic (MD) simulation their stability, interaction, and conformation were analyzed. In summary, this study identified the most suitable compounds for targeting Mpro, and we recommend using these compounds as potential drug molecules against SARS-CoV-2 after follow up studies.

## 1. Introduction

Reports suggest that the SARS-CoV-2 virus penetrates target tissues by manipulating two important proteins present on the surface of cells. The two key proteins are transmembrane serine protease 2 (TMPRSS2) and angiotensin-converting enzyme 2 (ACE2). The SARS-CoV-2 virus belongs to the β category of human coronaviruses [1,2,3], and its genomic organization is similar to that of other coronaviruses [4]. The viral genomic RNA (27–32 Kb) codes both structural and non-structural proteins. The structural proteins include membrane (M), envelope (E), nucleocapsid (N), hemagglutinin-esterase (HE), and spike (S) proteins. These proteins are known to facilitate the transmission and replication of viruses in host cells [5]. The replicase gene (ORF1a) and protease gene (ORF1b) encode polyprotein1a (pp1a) and polyprotein1ab (pp1ab). These polyproteins are further processed by Papain-like protease (PLpro) and Chymotrypsin-like protease (3CLpro) to generate nonstructural proteins (nsp) [3,6]. The main protease (M^pro^) is an essential enzyme, which plays a vital role in the lifecycle of the virus and can therefore be used in research efforts to identify potential target drugs. Additionally, since no proteases with M^pro^-like cleaving characteristics are found in humans, any potential protease inhibitors are likely to be nontoxic to humans.

The testing of broad-spectrum antiviral drugs is currently in process. However, despite unprecedented research efforts, efficient targeted therapies (which could provide a long-term solution to COVID-19) have still not been identified. Computer-aided drug discovery (CADD) methodologies have been widely used during the past decade and are a powerful tool to study protein-drug and protein-protein interactions. In recent developments, CADD methodologies are being used as a key resource for drug discovery to mitigate the COVID-19 pandemic [7,8,9]. Cava et al. have identified potential drug candidates that could impact the spread of COVID-19, such as: nimesulide, fluticasone propionate, and thiabendazole. Cava et al. used in silico gene-expression profiling to study the mechanisms of the ACE2 and its co-expressed genes [10]. Wang et al. conducted virtual screening of authorized drugs along with those that are in clinical trials to identify drug candidates against 3CLpro [11]. Liang et al., used molecular dynamics simulation to reveal the binding stability of an α-ketoamide inhibitor within the SARS-CoV-2 main protease (M^pro^) [12]. Gaudêncio and Florbela used CADD methodologies to screen natural marine products to identify effective ligands with SARS-CoV-2 main protease (M^pro^) with inhibiting potential [13]. Another potential approach is drug repurposing, which includes the screening of pre-existing drug compounds with anti-SARS-CoV-2 properties, which is followed by target identification and functional and structural characterization of any targeted enzymes. Finally, after successful screening and characterization, clinical trials can commence. In addition to the drug molecules, there are reports on applications of nanomaterials, such as metal-based, two-dimensional, and colloidal nanoparticles and nanomicelles, for antiviral and virus sensing applications [14,15,16,17]. Despite their small size and selective nature, nanoparticles have proved to be effective against wide range of pathogens, including bacteria and viruses. However, some metal-based nanoparticles have also been reported to have non-specific bacterial toxicity mechanisms, thereby reducing the chances of developing resistance as well as expanding the spectrum of antimicrobial activity [18]. Although the interest in designing nanomaterial-based, non-traditional drugs is growing, more advanced research is required to uncover their full potentials for being considered as promising agents against SARS-CoV-2.

To date, no specialized drugs are available on the market to cure COVID-19. Over recent years, the triazole group-based ligands have attracted the interest of the scientific community due to their comprehensive and multipurpose medicinal applications. Reports have been published stating that this group of ligands have potential antiviral, antibacterial, antifungal, antiparasitic and anti-inflammatory applications. Moreover, owing to the nature of their chemical properties, this group of ligands can be easily synthesized [19,20,21]. The triazole group-based ligands could be a potential drug-candidate for use against the SARS-CoV-2 virus [22,23]. Efforts to develop efficient therapeutic strategies against COVID-19 are still in progress.

In this study, we had evaluated the potential of the triazole ligands as effective antiviral agents. We identified the most suitable anti-SARS-CoV-2 candidate chemicals (based on their molecular docking scores), which were then further analyzed for positive ADMET properties. Scientists across the world are researching different antiviral compounds, to identify those with the highest potential effectivity against SARS-CoV-2 as well as having low or no toxicity for humans. Our results suggest that the recommended drugs in this study may be candidates for use in the treatment of COVID-19. Even though triazole ligands are already clinically approved drugs, they would still require clinical trials prior to repurposing as anti-COVID-19 medicines (Figure 1).

## 2. Results

### 2.1. Structural Analysis

The protein structure used for the molecular docking and simulation studies is shown in Figure 2. The binding pocket volume and surface area were determined through the CASTp webserver, utilizing previous findings [24]. A binding pocket was predicted at the surface as well as in the interior of proteins. The binding pocket volume of M^pro^ was 402.7 (SA) (Figure 3), which signifies an optimum space for ligand binding. All the participating residues are listed in Appendix A.

### 2.2. Molecular Docking

To identify a potential SARS-CoV-2 protease inhibitor, the structure-based molecular docking approach was performed on 171 triazole based compounds. These selected compounds have therapeutic potential against cancer, infectious diseases, and some other diseases. All 171 compounds were docked with the SARS-CoV-2 (M^pro^) chain A using target specific docking (pre-identified pocket with CastP). Out of 171 compounds, 27 compounds gave a docking score of −10.2 to −8 kcal/mol (Appendix A). The list of compounds, based on their binding energies (PyRx based Vina scores) of the highest ranked position of the docked ligand with SARS-CoV-2 main protease, are shown in Table 1 and Appendix A.

Four best ligand molecules were selected based on the top hit criteria and were further analyzed for molecular interactions with SARS-CoV-2 (M^pro^) (Table 1, Appendix A). The ligands are 1-{3,4-diazatricyclo[9.4.0.0^{2,7}]pentadeca-1(15),2(7),3,5,11,13-hexaen-5-yl}-*N*3-[(7S)-7-(pyrrolidin-1-yl)-6,7,8,9-tetrahydro-5Hbenzo[7]annulen-2-yl]-1*H*-1,2,4-triazole-3,5-diamine (Bemcentinib;DB12411), 2-(2*H*-1,2,3-benzotriazol-2-yl)-6-{[3-(2*H*-1,2,3-benzotriazol-2-yl)-2-hydroxy-5-(2,4,4-trimethylpentan-2-yl)phenyl]methyl}-4-(2,4,4-trimethylpentan-2-yl)phenol (Bisoctrizole;DB11262), (5-{3-[5-(Piperidin-1-Ylmethyl)-1h-Indol-2-Yl]-1h-Indazol-6-Yl}-2h-1,2,3-Triazol-4-Yl)methanol (PYIITM;DB07213), *N*-{3-[5-(1*H*-1,2,4-triazol-3-yl)-1*H*-indazol-3-yl]phenyl}furan-2-carboxamide (NIPFC;DB07020). Bemcentinib (DB12411 an investigational drug for the treatment of non-small cell lung cancer) (Appendix A) showed the highest binding energy, −10.2 kcal/mol, with the SARS-CoV-2 M^pro^ (Table 1). The results showed two hydrogen bonds with two main protease residues, Ser46, Thr26. Bemcentinib also showed one hydrophobic interaction (Pi-Alkyl) with Met49, residues of the SARS-CoV-2 M^pro^ enzyme (Figure 4, and Table 1).

In terms of highest binding energy, the other three potent organic triazole based compounds were Bisoctrizole (DB11262), PYIITM (DB07213), and NIPFC (DB07020) (Table 1, Appendix A). Bisoctrizole (DB11262 is a benzotriazole-based organic molecule that absorbs, reflects, and scatters both UV-A and UV-B rays) showed −9 kcal/mol binding energy against SARS-CoV-2 M^pro^ (Table 1). The interaction study showed two hydrogen bonds with M^pro^ residues, Cys44 and Gln189. Bisoctrizole also showed one unfavorable donor-donor interaction with residue Thr25 and one hydrophobic interaction (Pi-Alkyl) with Leu50 (Figure 4 and Table 1).

PYIITM (DB07213) showed −8.8 kcal/mol binding energy against SARS-CoV-2 M^pro^ (Table 1). The interaction study showed four hydrogen bonds with M^pro^ residues, three with His41, and one with Thr45, while PYIITM showed one electrostatic interaction (Pi Sigma) with residue Met49 and one hydrophobic interaction (Pi-Alkyl) with Cys44 (Figure 4 and Table 1).

NIPFC (DB07020) also showed −8.8 kcal/mol binding energy against SARS-CoV-2 M^pro^ (Table 1). The interaction study showed two hydrogen bonds with M^pro^ residues, Cys44 and Asn142, also on NIPFC, showed one hydrophobic interaction (Pi-Alkyl) with Met49 (Figure 4 and Table 1).

In our study, the ligands 11a and 11b (crystalized ligand structure used as inhibitor of M^pro^ in previous study) [25] were also docked against M^pro^ for assessment purposes. The 11a and 11b inhibitory ligands docking scores is low (−7.2 kcal/mol and −7.5 kcal/mol, Appendix A), whereas our best triazole ligands showed binding affinities of −10.2 kcal/mol (Bemcentinib (DB12411)), −9 kcal/mol (Bisoctrizole:DB11262), −8.8 kcal/mol (PYIITM:DB07213), and −8.8 kcal/mol (NIPFC:DB07020). A previous study suggests that 17 (Thr25, Thr26, His41, Cys44, Met49, Phe140, Asn142, Gly143, Cys145, His163, His164, Met165, Glu166, Pro168, Asp187, Arg188, Gln189) amino acids were participating or present in the M^pro^ and inhibitory ligands interaction [25]. Our protein–ligand interaction study suggested that seven amino acids (Thr25, Thr26, His41, Cys44, Met49, Asn142, Gln189) were involved in M^pro^ inhibition. Interestingly, these amino acids are also involved in M^pro^–Bemcentinib, M^pro^–Bisoctrizole, M^pro^–PYIITM, and M^pro^–NIPFC interaction, which indicates that all four triazole based ligands have binding affinity with amino acids, which play crucial roles in M^pro^ inhibition. In these terms, it can be concluded that Bemcentinib, Bisoctrizole, PYIITM, and NIPFC can be used as potential M^pro^ inhibitors.

### 2.3. Absorption, Distribution, Metabolism, Excretion, and Toxicity (ADMET) Analysis

Based on highest docking score, four ligands were selected for pharmacokinetics, including: the Lipinski rule of five, drug likeness, and ADMET analysis. Results obtained from the Lipinski rule of five are listed in Appendix A. PYIITM (DB07213) and NIPFC (DB07020) satisfied all the Lipinski rule parameters. Whereas the other two compounds, Bemcentinib (DB12411) and Bisoctrizole (DB11262), violated two Lipinski rules, previous studies suggested that, with two violations, compounds could be used as orally active antiviral agents [26]. However, all four compounds show favorable drug-likeness properties (Appendix A). ADMET properties of the four selected compounds were analyzed by a free pkCSM (http://biosig.unimelb.edu.au/pkcsm/prediction, accessed on 28 February 2021) web tool.

#### 2.3.1. Absorption

Drug absorption is mainly analyzed through the water solubility of compounds, cell permeability using colon carcinoma (Caco-2) cell line, human intestinal absorption, skin permeability, and whether the molecule is a P-glycoprotein substrate or inhibitor [27]. The compound water solubility reflects the compound solubility in water at 25 °C. All the selected compounds are moderately soluble in water (Table 2). Caco-2 cell permeability and human intestinal absorption determine the ultimate bioavailability; a drug having a value of more than 0.90 is considered readily permeable [26]. Bemcentinib (DB12411) showed particularly good permeability, whereas Bisoctrizole (DB11262) and PYIITM (DB07213) showed moderate permeability (Table 2), but NIPFC (DB07020) showed negligible permeability.

The human intestine is the primary site for drug absorption. A previous study suggested that a molecule with >30% absorbency is considered readily absorbed [27]. In silico absorbance analysis showed that Bemcentinib (DB12411) and Bisoctrizole (DB11262) have a 100% absorbance rate in the human intestine (Table 2), whereas the other compounds, PYIITM (DB07213) and NIPFC (DB07020), achieve a >80% absorbance rate. This clearly indicates that all the organic triazole based ligands have a high absorbance rate in the human intestine. All compounds were substrates for P-glycoprotein, except Bisoctrizole (DB11262). All four compounds were P-glycoprotein II inhibitors. Only Bemcentinib (DB12411) showed inhibition against P-glycoprotein I (Table 2).

#### 2.3.2. Distribution

The distribution was calculated using the following parameters: human volume of distribution, human fraction unbound in plasma, blood-brain barrier, and central nervous system permeability. In the bloodstream, drugs are generally transported in a free or unbound state or in a partly reversibly bound state. However, irrespective of the transportation state, the steady-state volume of distribution (VDss) remains one of the key pharmacokinetic parameters that must be considered when designing a drug dose range. VDss can be defined as the theoretical volume of a particular drug dose, which vary and give a similar blood plasma concentration. Generally, the greater the VDss value, the more a drug is distributed in tissue rather than plasma. However, for antibiotics and antivirals, more wide-ranging tissue distribution is desirable [27]. VDss is considered low if the log of the VDss value is lower than −0.15, while a value >0.45 is considered high [27]. Of the four compounds in question, Bemcentinib (DB12411) showed the highest distribution value, followed by PYIITM (DB07213) (Table 2). Bisoctrizole (DB11262) showed the lowest distribution value of the four compounds. The effectiveness of a drug may vary depending on the limit to which it can bind to blood proteins. The more effective the binding of the drug with blood proteins, the more efficiently the drug compounds can transverse the cellular membrane [27]. Fraction unbound to human plasma ranges between 0.02 to 1.0 [28]. All compounds showed a high fraction unbound value to human plasma, except NIPFC (DB07020) (Table 2).

#### 2.3.3. Metabolism

The metabolism of a drug depends upon the molecule being a Cytochrome P450 substrate or inhibitor. Bemcentinib (DB12411) showed moderate inhibition (CYP2C19, CYP3A4) of the cytochrome enzymes, whereas Bisoctrizole (DB11262) showed non-inhibitory properties against all enzymes (Table 3). PYIITM (DB07213) showed inhibition activity against only CYP1A2, whereas NIPFC (DB07020) showed inhibition against all cytochrome enzymes (Table 3). The results indicate that the Bisoctrizole (DB11262), PYIITM (DB07213), and Bemcentinib (DB12411) will be metabolized by the action of the cytochrome enzymes. On the other hand, NIPFC (DB07020) will not be metabolized by the cytochrome enzymes due to its inhibitory nature against all cytochrome enzymes.

#### 2.3.4. Excretion

Organic cation transporter 2 (OCT2) belongs to the category of renal uptake transporters, which are known to play important roles during deposition and clearing of drugs from the kidneys [28]. Excretion depends on factors such as total clearance and whether the molecule is a renal OCT2 substrate. None of the triazole compounds act as a substrate for Renal OCT2 and can be removed from the body through the renal system. Except PYIITM (DB07213), all the selected compounds show total clearance of less than log (CLtot) 1 mL/min/kg (Table 4).

#### 2.3.5. Toxicity

A negative AMES result indicates that the molecule is non-mutagenic and non-carcinogenic. None of the selected triazole compounds showed AMES toxicity except Bemcentinib (DB12411) (Table 4). Bemcentinib (DB12411) is under investigation as an anti-cancer drug against small lung tumors. The maximum recommended tolerance dose (MRTD) provides an estimate of the toxic dose in humans. MRTD values less than or equal to log 0.477 (mg/kg/day) is considered low [28]. Bemcentinib (DB12411) and Bisoctrizole (DB11262) had low toxicity to humans whereas PYIITM (DB07213) and NIPFC (DB07020) showed toxicity (Table 4). All four triazole compounds were not skin sensitive (Table 4). A molecule with a high oral rat acute toxicity (LD50) value is less lethal than the lower LD50 value [27,29]. For a given molecule, the LD50 is the amount that causes the death of 50% of the test animals [27,29]. All the selected ligands showed high oral rat acute toxicity (LD50) value (Table 4). The lethal concentration values (LC50) represent the concentration of a molecule necessary to cause 50% of fathead minnow death. For a given molecule, if the log LC50 < 0.5 mM (log LC50 < −0.3), then it is regarded as having high acute toxicity [29,30]. All three triazole compounds showed a satisfactory score that indicated that they are less toxic, except for Bisoctrizole (DB11262) (Table 4).

### 2.4. In Silico Antiviral Prediction

Bemcentinib showed more than 50.34% antiviral activity against all tested viruses, with 60.71% antiviral activity against HIV (Appendix A); Bisoctriazole showed more than 61.38% antiviral activity against all tested viruses, with more than 60.32% activity against HIV; and PYIITM showed more than 62.49% antiviral activity against all tested viruses, with 48.11% antiviral activity against HIV. NIPFC showed more than 36% antiviral activity against all tested viruses, with 60.61% antiviral activity against HIV (Appendix A). Based on antiviral prediction, it can be concluded that Bemcentinib, Bisoctriazole, and PYIITM can be used as potent antiviral drugs against the SARS-CoV-2 virus (Appendix A), because previous case and clinical studies suggested that some antiviral drugs mostly used for HIV showed effects against SARS-CoV-2 virus [31,32].

#### 2.4.1. MD Simulation and Analysis

Based on the best docking score four top hit molecules, Bemcentinib (−10.2 kcal/mol), Bisoctriazole (−9 kcal/mol), PYIITM (DB07213) (−8.8 kcal/mol), and NIPFC (DB07020) (−8.8 kcal/mol) were selected for MD simulation studies (with all-atoms). The dynamic features of the protease-inhibitor interactions were analyzed based on various parameters, such as RMSD, RMSF, Rg, H-bonds, SASA, and interaction energy.

#### 2.4.2. RMSD Analysis

To determine M^pro^ docked complex conformation stability with drug compounds, Bemcentinib (−10.2 kcal/mol), Bisoctriazole (−9 kcal/mol), PYIITM (−8.8 kcal/mol), and NIPFC (DB07020), the backbone root mean square deviation (Cα-RMSD) were computed, as shown in Figure 5. The result shows that the RMSD trajectory of M^pro^–Bemcentinib was equilibrated during 0–5 ns and remained steady with a RMSD value ∼2.0 ± 0.2 Å at the end of simulation at 40 ns (Figure 5A), which indicates very stable structural complexity of the M^pro^–Bemcentinib complex. Likewise, the RMSD plot of the M^pro^–Bisoctriazole complex showed a reasonably stable structure during the 40 ns stimulation process. Mpro–Bisoctriazole complex exhibited RMSD ∼1.7 Å (Figure 5A). Similarly, M^pro^–PYIITM and M^pro^–NIPFC RMSD plots showed RMSD values ∼1.6 Å and ∼1.75 Å, respectively, which clearly indicates the structural stability of M^pro^–PYIITM and M^pro^–NIPFC complexes. (Figure 5A). All the RMSD values indicate a very stable structural conformation of the M^pro^ protein with all four ligand compounds.

#### 2.4.3. Rg Analysis

Additionally, the conformation stability of the M^pro^–ligand was evaluated by the radius of gyration (Rg). The Rg parameter is used by computational biologists to describe the structural compactness of proteins. To examine the structural compactness and integrity of M^pro^–ligand bound complexes, the radius of gyration (Rg) is calculated for each system [33,34]. From Figure 5, it can be observed that the structure of M^pro^–Bemcentinib, M^pro^–Bisoctriazole, M^pro^–PYIITM, and M^pro^–NIPFC stabilized around an Rg value 22.5 Å ± 0.1 Å, and it can be seen that there was no structural drift (Figure 5B). The structural compactness of M^pro^–drug complexes calculated by Rg analyses suggested stable molecular interaction with all four compounds, which are stabilized in 22.5 Å ± 0.1 Å (Figure 5B).

#### 2.4.4. RMSF Analysis

The RMSF plots of M^pro^–Bemcentinib, M^pro^–Bisoctriazole, M^pro^–PYIITM, and M^pro^–NIPFC represent that the amino acid residues belonging to termini (N-and C-terminal) and loops have an average atomic fluctuation >1.5 Å (Figure 5C). In divergence, the conformational dynamics of stable secondary structure, α-helices, and β-sheets (interacting protein residues with the ligand compounds) remain stable during the whole simulation process, providing an indication of the stability of molecular interactions of M^pro^ with triazole based ligand compounds. The average atomic fluctuations were measured using RMSF plots, which suggested that all four M^pro^–drug complexes showed similar 3D binding patterns, which clearly indicates that all four triazole based compounds were well accommodated at the binding pocket of M^pro^ with favorable molecular interactions.

#### 2.4.5. H-Bonds Analysis

Furthermore, the time evolution plot of hydrogen bond occupancy (H-bonds) between target SARS-CoV-2 main protease and inhibitors was computed. H-bonds are also designated as the “master key of molecular recognition” due their crucial role in ligand binding and enzyme catalysis. Although H-bonds are weaker bonds compared to covalent bonds, their flexibility makes them the most important physical interaction in systems of bio-compounds in aqueous solution. They are critical for maintaining the shape and stability of protein structure. In the case of M^pro^–Bemcentinib interactions, initially, four H-bonds were detected; however, over time, the number of H-bonds reduced. No H-bonds were obtained from approximately 24–32 ns. After this time, some spikes for H-bonds were identified. Finally, at 40 ns, one H-bond was detected, which came close to supporting our docking interaction data. In the case of M^pro^–Bisoctriazole, initially, four H-bonds were detected; thereafter, the number of H-bonds varied from two to three, which strongly supports our docking calculations. In the case of PYIITM and M^pro^, we detected four to five H-bonds, and NIPFC maintained two hydrogen bonds throughout the simulation time, which strongly agreed with our docking interaction calculations (Figure 5D).

#### 2.4.6. SASA Analysis

Hydrophobic interactions can be considered determinants of protein conformational dynamics. Protein conformational dynamics are known to guarantee the structural stability of molecular interactions [34,35]. Computation of the solvent-accessible surface area (SASA) is an important parameter when studying changes in structural features of M^pro^–Bemcentinib, M^pro^–Bisoctriazole, M^pro^–PYIITM, and M^pro^–NIPFC complexes. The proper functioning of protein–ligand complexes depend on how well the protein maintains its fold during the interactions. Figure 5E (black line) shows that the complex structure SARS-CoV-2 M^pro^ occupied with the Bemcentinib had an average SASA value of 166.25 nm^2^ ± 2 nm^2^. The complex structures SARS-CoV-2 M^pro^ occupied with Bisoctriazole, PYIITM, and NIPFC had an average SASA value of 168.50 nm^2^ ± 2 nm^2^ (Figure 5E red, gree, blue line). Almost no change in orientation in the protein surface was detected for the molecular interaction of SARS-CoV-2 M^pro^ with Bisoctriazole, PYIITM, and NIPFC. However, in the case of interaction with Bemcentinib, a negligible decrease in the protein accessible area was detected, which is an indication of insignificant orientational change in the protein surface. Thus, the SASA investigation for all four complexes suggested no significant changes in the conformational dynamics of M^pro^–Bemcentinib, M^pro^–Bisoctriazole, M^pro^–PYIITM and M^pro^–NIPFC complexes.

#### 2.4.7. Interaction Energy Analysis

The short-range electrostatic (Coul-SR) and van der Waals/hydrophobic (LJ-SR) interaction energies between M^pro^–Bemcentinib, M^pro^–Bisoctriazole, M^pro^–PYIITM, and M^pro^–NIPFC complexes explained promising electrostatic as well as hydrophobic interactions. For M^pro^–Bemcentinib, average values of Coul-SR, −7.19 ± 3.2 kJ/mol, and LJ-SR, −109.162 ± 4.9 kJ/mol, were observed. For M^pro^–Bisoctriazole, a Coul-SR of −25.37 ± 4 kJ/mol and an LJ-SR of −67.22 ± 6.1 kJ/mol were observed. M^pro^–PYIITM complex exerts a Coul-SR of −61.02 ± 6.3 kJ/mol and an LJ-SR of −94.07 ± 1.3 kJ/mol. M^pro^–NIPFC complexes showed a Coul-SR of −11.21 ± 5.4 kJ/mol and an LJ-SR of −30.76 ± 1.2 kJ/mol (Figure 5F). This suggested that the role of hydrophobic interaction was more important than the electrostatic interactions [36] in stabilizing the complex, a conclusion that is also supported by previous experimental data.

## 3. Materials and Methods

### 3.1. Target and Ligand Preparation

The crystal structure of SARS-CoV-2 main protease in complex with an inhibitor 11b (PDB-ID: 6M0K at resolution 1.80 Å, R-Value Free: 0.193, R-Value Work: 0.179 and R-Value Observed: 0.180) was retrieved from RCSB PDB database (http://www.rcsb.org/pdb, accessed on 27 February 2021) and used in the present study. The inhibitor 11b was removed from the structure with Chimera 1.15 for docking studies. The 3D SDF structure library of 171 triazole based compounds was downloaded from the DrugBank 3.0 database (https://go.drugbank.com/; accessed on 27 January 2021). All compounds were then imported into Open Babel software (Open Babel development team, Cambridge, UK) using the PyRx Tool and were exposed to energy minimization. The energy minimization was accomplished with the universal force field (UFF) using the conjugate gradient algorithm. The minimization was set at an energy difference of less than 0.1 kcal/mol. The structures were further converted to the PDBQT format for docking.

### 3.2. Protein Pocket Analysis

The active sites of the receptor were predicted using CASTp (http://sts.bioe.uic.edu/castp/index.html?2pk9, accessed on 28 January 2021). The possible ligand-binding pockets that were solvent accessible, were ranked based on area and volume [37].

### 3.3. Molecular Docking and Interaction Analysis

AutoDock Vina 1.1.2 in PyRx 0.8 software (ver.0.8, Scripps Research, La Jolla, CA, USA) was used to predict the protein-ligand interactions of the triazole compounds against the SARS-CoV-2 main protease protein. Water compounds and attached ligands were eliminated from the protein structure prior to the docking experiments. The protein and ligand files were loaded to PyRx as macromolecules and ligands, which were then converted to PDBQT files for docking. These files were similar to pdb, with an inclusion of partial atomic charges (Q) and atom types (T) for each ligand. The binding pocket ranked first was selected (predicted from CASTp). Note that the other predicted pockets were relatively small and had lesser binding residues. The active sites of the receptor compounds were selected and were enclosed within a three-dimensional affinity grid box. The grid box was centered to cover the active site residues, with dimensions x = −13.83 Å, y = 12.30 Å, z = 72.67Å. The size of the grid wherein all the binding residues fit had the dimensions of x = 18.22 Å, y = 28.11 Å, z = 22.65 Å. This was followed by the molecular interaction process initiated via AutoDock Vina from PyRx [38]. The exhaustiveness of each of the three proteins was set at eight. Nine poses were predicted for each ligand with the spike protein. The binding energies of nine docked conformations of each ligand against the protein were recorded using Microsoft Excel (Office Version, Microsoft Corporation, Redmond, Washington, USA). Molecular docking was performed using the PyRx 0.8 AutoDock Vina module. The search space included the entire 3D structure chain A. Protein-ligand docking was initially visualized and analyzed by Chimera 1.15. The follow-up detailed analysis of amino acid and ligand interaction was performed with BIOVIA Discovery Studio Visualizer (BIOVIA, San Diego, CA, USA). The compounds with the best binding affinity values, targeting the COVID-19 main protease, were selected for further molecular dynamics simulation analysis.

### 3.4. Absorption, Distribution, Metabolism, Excretion, and Toxicity (ADMET) Analysis

Pharmacokinetic parameters related to the absorption, distribution, metabolism, excretion, and toxicity (ADMET) play a substantial role in the detection of novel drug candidates. To predict candidate molecules using in silico methods pkCSM (http://biosig.unimelb.edu.au/pkcsm/prediction, accessed on 28 February 2021), webtools were used. Parameters such as AMES toxicity, maximum tolerated dose (human), hERG I and hERG II inhibitory effects, oral rat acute and chronic toxicities, hepatotoxicity, skin sensitization, and T. pyriformis toxicity and fathead minnow toxicity were explored. In addition to these, molecular weight, hydrogen bond acceptor, hydrogen bond donor, number of rotatable bonds, topological polar surface area, octanol/water partition coefficient, aqueous solubility scale, blood-brain barrier permeability, CYP2D6 inhibitor hepatotoxicity, and number of violations of Lipinski’s rule of five were also surveyed.

### 3.5. In Silico Antiviral Assay

A quantitative structure-activity relationship (QSAR) approach was used in AVCpred to predict the antiviral potential of the candidates through the AVCpred server (http://crdd.osdd.net/servers/avcpred/batch.php, accessed on 28 January 2021). This prediction was conducted based on the relationships connecting molecular descriptors and inhibition. In this method, we used the most promising compounds screened against: human immunodeficiency virus (HIV), hepatitis C virus (HCV), hepatitis B virus (HBV), human herpesvirus (HHV), and 26 other important viruses (listed in Appendix A), with experimentally validated percentage inhibition from ChEMBL, a large-scale bioactivity database for drug discovery. This was followed by descriptor calculation and selection of the best performing molecular descriptors. The latter were then used as input for a support vector machine (in regression mode) to develop QSAR models for different viruses, as well as a general model for other viruses. [39].

### 3.6. MD Simulation Studies

The five best protein-ligand complexes were chosen for MD simulation according to the lowest binding energy with the best docked pose. Additional binding interactions were used for molecular simulation studies. The simulation was carried out using the GROMACS 2020 package (University of Groningen, Groningen, Netherland), utilizing a charmm36 all-atom force field using empirical, semi-empirical and quantum mechanical energy functions for molecular systems. The topology and parameter files for the input ligand file were generated on the CGenff server (http://kenno.org/pro/cgenff/, accessed on 27 February 2021). A TIP3P water model was used to incorporate the solvent, adding counter ions to neutralize the system. The energy minimization process involved 50,000 steps for each steepest descent, followed by conjugant gradients. PBC condition was defined for x, y, and z directions, and simulations were performed at a physiological temperature of 300 K. The SHAKE algorithm was applied to constrain all bonding involved, hydrogen, and long-range electrostatic forces treated with PME (particle mesh Ewald). The system was then heated gradually at 300 K, using 100 ps in the canonical ensemble (NVT) MD with 2 fs time step. For the isothermal-isobaric ensemble (NPT) MD, the atoms were relaxed at 300 K and 1 atm using 100 ps with 2 fs time step. After equilibrating the system at desired temperature and pressure, the MD run for the system was carried out at 40 ns with time step of 2 fs at 20,000,000 steps. The coordinates and energies were saved at every 10 ps for analysis.

MD simulation trajectories were analyzed by using a trajectory analysis module integrated into the GROMACS 2020.01 simulation package, qtgrace, VMD, and Chimera software (University of California San Francisco, San Francisco, CA, USA). The trajectory files were first analyzed using GROMCAS tools: gmx rmsd, gmx gyrate, gmx sasa, gmx hbond, gmx covar, and gmx energy for extracting the graph of root-mean square deviation (RMSD), root-mean square fluctuations (RMSFs), radius of gyration (Rg), solvent accessible surface area (SASA), hydrogen bond, principal component, potential energy, kinetic energy, and enthalpy, with python3 free energy surface calculation and visualization. The .mdp files scripts for NVT, NPT, MD production and interaction energy were added in the Appendix A.

## 4. Conclusions

The present study explored the molecular interactions of ligands, Bemcentinib, Bisoctriazole, PYIITM, and NIPFC. These were analyzed as prospective drug candidates against the SARS-CoV-2 (M^pro^) protein. The screened compounds showed excellent docking scores, excellent pharmacokinetic profiles, MD simulation data, and interaction energy profile. Furthermore, these compounds positively cohere with the predetermined amino acid residues present in the core palm region of the M^pro^ protein, thus inhibiting the processing of the polyproteins that are translated from viral RNA. The ADMET results revealed excellent bioavailability and enzymatic inhibitory effects. The four compounds under investigation in this paper are already approved for other medical applications. This paper demonstrated the first occasion that the inhibitory action of these compounds was simulated for use against the SARS-CoV-2 virus. The interaction energy estimation using GROMACS extension revealed that the selected inhibitors, Bemcentinib, Bisoctriazole, PYIITM, and NIPFC, possess extremely high interaction energy and molecular affinity. Therefore, we propose that the selected compounds might be used as lead compounds in COVID-19 therapy. The pharmacological profiling, docking analysis, MD simulation, MD trajectory, and interaction energy studies indicated that Bemcentinib, Bisoctriazole, PYIITM, and NIPFC could be used as possible drug candidates for inhibition against the SARS-CoV-2 M^pro^ protein to interrupt the essential role it plays in processing polyproteins translated from viral RNA. Based on the data presented in this paper, the compounds investigated in this study could be considered for further clinical studies and thereafter for potential treatment of COVID-19.

## Figures and Tables

**Figure 1 molecules-26-06199-f001:**
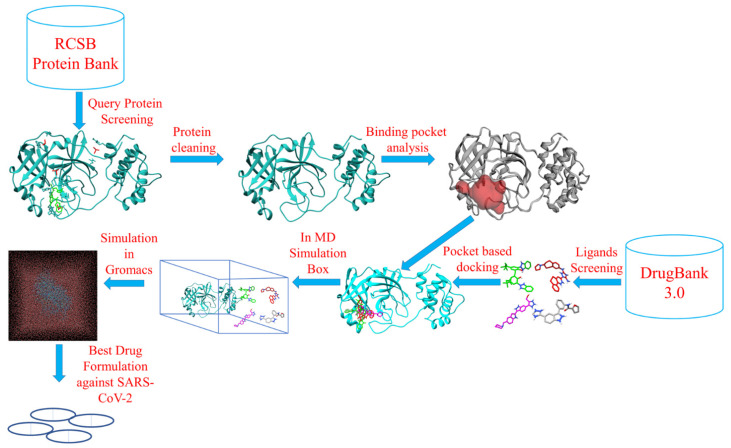
Schematic diagram of the workflow.

**Figure 2 molecules-26-06199-f002:**
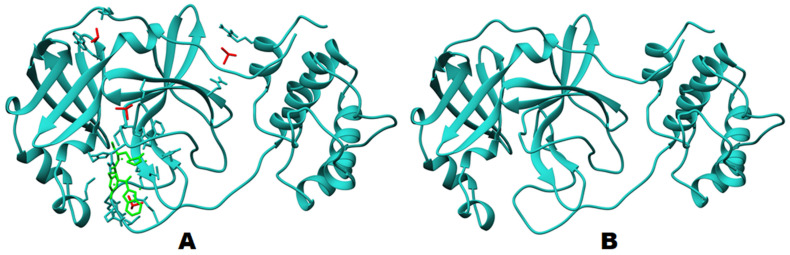
Protein structures: (**A**). before docking studies and (**B**). after cleaning of ligand and additional molecules, used for further docking and MD simulation.

**Figure 3 molecules-26-06199-f003:**
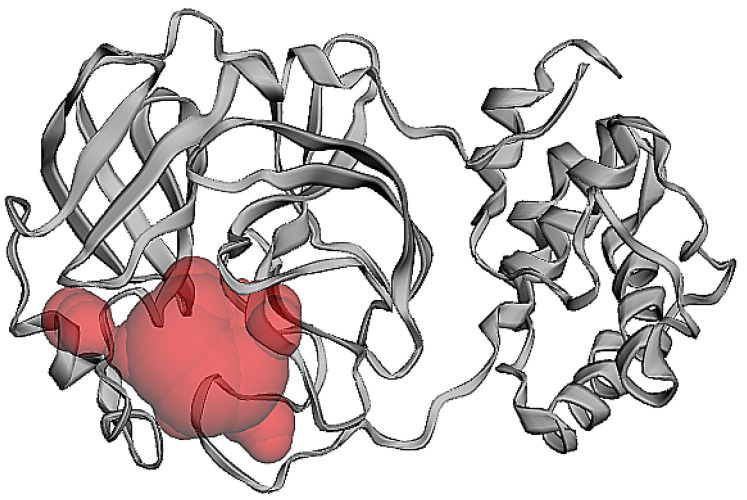
Binding pocket analysis (predicted by CASTp software).

**Figure 4 molecules-26-06199-f004:**
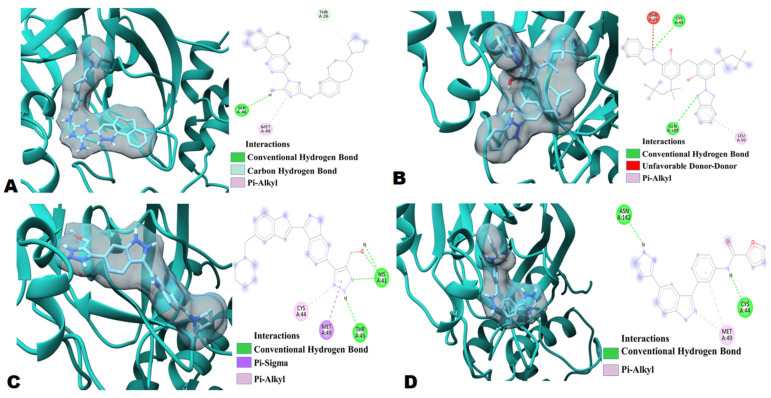
Molecular docking analysis of M^pro^ system in complex with (**A**). Bemcentinib (**B**). Bisoctriazole (**C**). PYIITM and (**D**). NIPFC.

**Figure 5 molecules-26-06199-f005:**
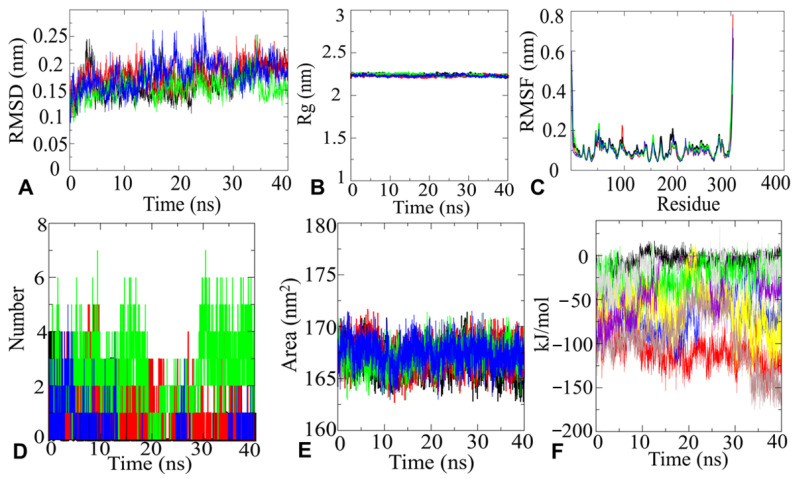
(**A**). RMSD plot of the M^pro^ system in complex with Bemcentinib, Bisoctriazole, PYIITM, and NIPFC. Here, black line defines Bemcentinib, red line defines Bisoctriazole, green line defines PYIITM, and blue line defines NIPFC. (**B**). Rg plot of the M^pro^ system in complex with Bemcentinib, Bisoctriazole, PYIITM, and NIPFC, which clearly indicates the compactness of the protein in the complex with ligand compounds. Here, black line defines Bemcentinib, red line defines Bisoctriazole, green line defines PYIITM, and blue line defines NIPFC. (**C**). RMSF analysis plot for SARS-CoV-2 main protease system in complex with Bemcentinib, Bisoctriazole, PYIITM, and NIPFC. Here, black line defines Bemcentinib, red line defines Bisoctriazole, green line defines PYIITM, and blue line defines NIPFC. (**D**). Hydrogen bond dynamics between SARS-CoV-2 Mpro in complex with Bemcentinib, Bisoctriazole, PYIITM, and NIPFC. Here, black line defines Bemcentinib, red line defines Bisoctriazole, green line defines PYIITM, and blue line defines NIPFC. (**E**). SASA plot for SARS-CoV-2 main protease system in complex with Bemcentinib, Bisoctriazole, PYIITM, and NIPFC. Here, black line defines Bemcentinib, red line defines Bisoctriazole, green line defines PYIITM, and blue line defines NIPFC. (**F**). Interaction energy plot for SARS-CoV-2 main protease system in complex with Bemcentinib, Bisoctriazole, PYIITM, and NIPFC. Here, black line defines Bemcentinib, red line defines Bisoctriazole, green line defines PYIITM, and blue line defines NIPFC.

**Table 1 molecules-26-06199-t001:** Organic triazole compounds used for further analysis for molecular interactions in the SARS-CoV-2 main protease.

Triazole Based Compounds	Binding Affinity Values (kcal/mol)	No. of H-bonds	H-bonds and Interacting Residues	No. of Other Interactions	Other Interaction and Interacting Residues
Bemcentinib (DB12411)	−10.2	2	Ser46, Thr26	1	Met49
Bisoctrizole (DB11262)	−9.0	2	Cys44, Gln189	1	Leu50
PYIITM (DB07213)	−8.8	4	His41 (3), Thr45 (1)	2	Met49, Cys44
NIPFC (DB07020)	−8.8	2	Cys44, Asn142	1	Met49

**Table 2 molecules-26-06199-t002:** ADMET pharmacokinetics; absorbance and distribution parameters.

Compounds/ Ligands	Water Solubility log mol/L	Caco-2 Permeability log 10^−6^ cm/s	Human Intestinal Absorption (%)	P-glycoprotein Substrate	P-glycoprotein I Inhibitor	P-glycoprotein II Inhibitor	VDss (log L/kg)	Fraction Unbound (Human)
Bemcentinib	−3.166	1.336	100	Yes	Yes	Yes	0.755	0.179
Bisoctrizole	−2.929	1.489	100	No	No	Yes	−1.227	0.437
PYIITM	−2.889	0.877	80.603	Yes	No	Yes	−0.083	0.161
NIPFC	−2.871	0.355	84.718	Yes	No	Yes	−0.557	−0.557

**Table 3 molecules-26-06199-t003:** ADMET pharmacokinetics; metabolism and excretion parameters.

Compounds/ Ligands	CYP2D6 Substrate	CYP3A4 Substrate	CYP1A2 Inhibitor	CYP2C19 Inhibitor	CYP2C9 Inhibitor	CYP2D6 Inhibitor	CYP3A4 Inhibitor
Bemcentinib (DB12411)	No	Yes	No	Yes	No	No	Yes
Bisoctrizole (DB11262)	No	Yes	No	No	No	No	No
PYIITM (DB07213)	Yes	Yes	Yes	No	No	No	No
NIPFC (DB07020)	Yes	Yes	Yes	Yes	Yes	Yes	Yes

**Table 4 molecules-26-06199-t004:** ADMET pharmacokinetics; toxicity parameters.

Compounds/ Ligands	AMES Toxicity	Total Clearance log ml/ min/kg	Renal OCT2 Substrate	Max. Tolerated Dose (Human)	Oral Rat Acute Toxicity (LD50)	Skin Sensitization	Minnow Toxicity
Bemcentinib (DB12411)	Yes	0.920	No	0.181	2.995	No	1.920
Bisoctrizole (DB11262)	No	−1.185	No	0.429	3.115	No	−5.882
PYIITM (DB07213)	No	1.088	No	0.529	2.517	No	1.985
NIPFC (DB07020)	No	0.305	No	0.602	2.890	No	3.334

## Data Availability

Not applicable.

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
