# Peer review of "In Silico Identification and Validation of Organic Triazole Based Ligands as Potential Inhibitory Drug Compounds of SARS-CoV-2 Main Protease"

_molecules, 2021, doi:10.3390/molecules26206199_

Round 1
Reviewer 1 Report
Comments: The authors evaluate the potential of the triazole ligands as effective antiviral agents and then further analyzed them for positive ADMET properties. This approach is straightforward, efficient, and possibly applicable to other virus systems. However, there are a few errors/concerns that should be addressed before considering publication. Hope the authors reflect on the following comments and improve the work.
- There are a number of typos across the manuscript and I hope that authors can carefully review the manuscript. For example, Page 4 Table 1, “No of H-bonds” should be “No. of H-bonds”.
- In Page 5 Figure 3D, the chemical drawing is blur and too small. Please kindly correct it.
- Figure 6 and Figure 7 are not consistent with regard to their font and the unit of x-axis (ns or ps). Please correct them.
- In the Introduction, the authors should provide sufficient background knowledge for antiviral and antibacterial applications. For example, in addition to pharmaceutical chemicals, please include other materials such as nanomaterials been used for antiviral and virus sensing applications.
- "Preparation of New Sargassum fusiforme Polysaccharide Long-Chain Alkyl Group Nanomicelles and Their Antiviral Properties against ALV-J." Molecules11 (2021): 3265.
- "Colloidal nanoparticle inks for printing functional devices: emerging trends and future prospects." Journal of Materials Chemistry A41 (2019): 23301-23336.
- "Engineered two-dimensional nanomaterials: an emerging paradigm for water purification and monitoring." Materials Horizons3 (2021): 758-802.
- "Enhancement of Biological and Pharmacological Properties of an Encapsulated Polyphenol: Curcumin." Molecules14 (2021): 4244.
Author Response
Comments: The authors evaluate the potential of the triazole ligands as effective antiviral agents and then further analyzed them for positive ADMET properties. This approach is straightforward, efficient, and possibly applicable to other virus systems. However, there are a few errors/concerns that should be addressed before considering publication. Hope the authors reflect on the following comments and improve the work.
We thank to the reviewer for taking time to review our manuscript. The comments provided by the respected reviewer will indeed improve the quality of the manuscript. We had responded to the raised questions in point-by-point manner, for your kind consideration. We had also highlighted the changes using the “Track Changes” function, as per journal requirement.
- There are a number of typos across the manuscript and I hope that authors can carefully review the manuscript. For example, Page 4 Table 1, “No of H-bonds” should be “No. of H-bonds”.
We corrected the typo errors in Table 1 and also throughout the manuscript.
- In Page 5 Figure 3D, the chemical drawing is blur and too small. Please kindly correct it.
We had improved the figure quality. The updated new figure is inserted in the text and submitted.
- Figure 6 and Figure 7 are not consistent with regard to their font and the unit of x-axis (ns or ps). Please correct them.
The figure quality was improved as per reviewer’s suggestion. Please see the new figures in the text and submitted.
- In the Introduction, the authors should provide sufficient background knowledge for antiviral and antibacterial applications. For example, in addition to pharmaceutical chemicals, please include other materials such as nanomaterials been used for antiviral and virus sensing applications.
- "Preparation of New Sargassum fusiforme Polysaccharide Long-Chain Alkyl Group Nanomicelles and Their Antiviral Properties against ALV-J." Molecules11 (2021): 3265.
- "Colloidal nanoparticle inks for printing functional devices: emerging trends and future prospects." Journal of Materials Chemistry A41 (2019): 23301-23336.
- "Engineered two-dimensional nanomaterials: an emerging paradigm for water purification and monitoring." Materials Horizons3 (2021): 758-802.
- "Enhancement of Biological and Pharmacological Properties of an Encapsulated Polyphenol: Curcumin." Molecules14 (2021): 4244.
We thank again to the reviewer for the comments and we had included the reviewer’s suggestions in the manuscript text. The newly incorporated lines are as follow:
“In addition to the drug molecules, there are also reports on application of nanomaterials like metal based, two-dimensional and colloidal nanoparticles, nanomicelles, for anti-viral and virus sensing applications [14-17]. Despite of their small size and selective nature nanoparticles have proved to be effective against wide range of pathogens including bacteria and viruses. However, some metal-based nanoparticles have also been reported to have non-specific bacterial toxicity mechanisms, thereby reducing the chances of developing resistance as well as expanding the spectrum of antimicrobial activity [18]. Although the interest for designing nanomaterials based-non-traditional drugs is growing, more advanced researches on these new incentives must increase to uncover their full potentials as they might be considered as promising agents against SARS-CoV-2.”

Reviewer 2 Report
In attention of the manuscript authors,
In the “molecules-1396478” manuscript, the author has made substantial research efforts to investigate the inhibiting potentialities of triazole-based compounds against the SARS-CoV-2 main protease (Mpro). In this regard, molecular docking, ADMET properties, antiviral predictions, and MD simulations, were tested on 171 candidates collected from the DrugBank database. As a final result, four candidates (Bemcentinib, Bisoctrizole, PYIITM, and NIPFC) presented high binding affinity values and potential to interrupt the main protease (Mpro) activities of the SARS-CoV-2 virus.
The outcomes provided by the manuscript, if written and interpreted in a clear manner, and enhanced by supplementary information based on the referee’s recommendations, could be a real gain for researchers interested in developing new triazole-based candidates capable to combat the SARS-CoV-2 virus.
In this context, the potential impact of the manuscript results in the research world, and with all due respect to the author’s work, the manuscript may be considered for publication in Molecules journal, after a major revision will be made to improve the manuscript.
The comments addressed to the manuscript authors.
- I recommend to draw a workflow to make it easier to follow the article, and also to read, check, and correct English throughout the manuscript.
- Please rewrite the following sentence from the abstract, in a clear way: “ In summary, the most suitable compounds successfully identified for targeting SARS-CoV-2 main protease (Mpro) and we can suggest that the identified compounds can be used for further experimental approach as potential drug molecules against SARS-CoV-2.”
- For text consistency, please write the names of the compounds with the same typeface (lowercase or uppercase). e.g. lines 118-119, IUPAC name of PYIITM;DB07213
- It is not very clear the reason for choosing only 4 out of 27 compounds to be further investigated. Just the fact that they presented the best docking score is not enough? Best docking score compared with what score? Why compounds with a score of -8.7 or -8.6…. -8.0 were not considered; the difference is extremely small compared to -8.8? Analyzing the binding affinities listed in Table S3, I could say that all compounds could be further investigated because you don’t have a score reference. It is not enough to select compounds arbitrary, only by score, without having a reference to compare!! The X-ray ligand should have displayed the reference to compare scores and finally select compounds!
- Section 2.2. Molecular docking – the authors present the same information both in a detailed discussion of section 2.2 and Table 1 without indicating significant information on the investigated compounds. I think it was very useful if the authors briefly presented the amino acids known to be significant for SARS-CoV-2 inhibitory activity and then compare the interactions made by their compounds (organic triazole) with the known interaction ones. Moreover, the authors should have compared the interactions made by those four compounds with that of the x-ray ligand (6MOK).
- I don't understand exactly, why did the authors predict the binding pocket volume and surface area if they already have a ligand with the proper binding site?
- Moreover, a final conclusion of section 2.2 is missing. The authors, as I mentioned, only list some remarks.
- I strongly suggest to rewrite this section by taken to account the above comments.
- Figures 3 and 4 are not relevant. The chemical structures of the selected compounds could be integrated into Table 1 at the appropriate names otherwise it just takes up space in the manuscript. Please change accordingly.
- Figure 5 indicates clearly and visibly the interaction types. However, if the author's considered Figure 4 important, then mix Figure 4 and 5 into one. In this way, both the interactions and the binding orientation of the compounds in the active site are preserved.
- The authors must be very careful about what they write. They show at the beginning of section 2.5 that, based on docking and pharmacokinetic analyzes selected four potential inhibitors of the SARS-CoV-2 protease! THIS IS COMPLETELY UNTRUE! The authors selected these four compounds after docking and then performed pharmacokinetic and MD analyses on them. So, these compounds have been selected before!!!! If not, the authors are invited to evaluate all 27 compounds by docking and pharmacokinetics and then decide the best compounds with possible antiviral activity and finally, to run MD. Correct accordingly!!
- MD Simulation - the authors discussed the H-bonds observed for Bemcentinib, Bisoctriazole, and PYIITM complexes during the 40s, but nothing about the NIPFC complex (just a picture)? For the latter, no interaction was observed during the 40ns?
- For MD simulation, the authors presented a great number of pictures which loads the article a lot? The same observation for the number of tables in section 2.3. I strongly recommend for the MD section that information such as RMDS or RMSF, or Rg, etc of all four compounds be plotted together. For example, when all 4 RMSFs are overlapped in a single graph, it may be easier to analyze. The same for RMSD or Rg, etc.
- The authors mentioned in Conclusions only 3 out of 4 compounds as potential SARs-COV-2 candidates. Why the authors excluded NIPFC compounds? On the contrary, if it is a compound that should be predicted with caution, it is Bemcentinib (see MD evaluation – some of the H-bond interaction have disappeared compared to the docking)
Conclusions sentence from the manuscript: "The pharmacological profiling, docking analysis, MD simulation, MD trajectory, and interaction energy studies indicated that Bemcentinib, 484 Bisoctriazole, and PYIITM could be used as possible drug candidates for inhibition against the SARS-CoV-2 Mpro protein to interrupt the essential role it plays in processing polyproteins translated from viral RNA.”
Author Response
In the “molecules-1396478” manuscript, the author has made substantial research efforts to investigate the inhibiting potentialities of triazole-based compounds against the SARS-CoV-2 main protease (Mpro). In this regard, molecular docking, ADMET properties, antiviral predictions, and MD simulations, were tested on 171 candidates collected from the DrugBank database. As a final result, four candidates (Bemcentinib, Bisoctrizole, PYIITM, and NIPFC) presented high binding affinity values and potential to interrupt the main protease (Mpro) activities of the SARS-CoV-2 virus.
The outcomes provided by the manuscript, if written and interpreted in a clear manner, and enhanced by supplementary information based on the referee’s recommendations, could be a real gain for researchers interested in developing new triazole-based candidates capable to combat the SARS-CoV-2 virus.
In this context, the potential impact of the manuscript results in the research world, and with all due respect to the author’s work, the manuscript may be considered for publication in Molecules journal, after a major revision will be made to improve the manuscript.
We thank to the reviewer for carefully reviewing our manuscript. We had responded to the raised questions in point-by-point manner, for your kind consideration. Additionally, we had checked the corrected the English language of our manuscript. We had also highlighted the changes using the “Track of Changes” function, as per journal requirement.
The comments addressed to the manuscript authors.
- I recommend to draw a workflow to make it easier to follow the article, and also to read, check, and correct English throughout the manuscript.
According to the reviewer’s suggestion a clear work flow is added in the main manuscript file, and the English is also corrected according to the reviewer’s suggestions by the native speaker.
- Please rewrite the following sentence from the abstract, in a clear way: “ In summary, the most suitable compounds successfully identified for targeting SARS-CoV-2 main protease (Mpro) and we can suggest that the identified compounds can be used for further experimental approach as potential drug molecules against SARS-CoV-2.”
This section is now modified according to the reviewer’s suggestion. According to the reviewer’s suggestion the this part is now modified.. The newly incorporated lines are as follow:
“In summary, this study identified the most suitable compounds for targeting Mpro and we recommend using these compounds as potential drug molecules against SARS-CoV-2, after ensuing some follow up studies”
- For text consistency, please write the names of the compounds with the same typeface (lowercase or uppercase). e.g. lines 118-119, IUPAC name of PYIITM;DB07213
This part is corrected now according to the reviewer’s suggestion.
- It is not very clear the reason for choosing only 4 out of 27 compounds to be further investigated. Just the fact that they presented the best docking score is not enough? Best docking score compared with what score? Why compounds with a score of -8.7 or -8.6…. -8.0 were not considered; the difference is extremely small compared to -8.8? Analyzing the binding affinities listed in Table S3, I could say that all compounds could be further investigated because you don’t have a score reference. It is not enough to select compounds arbitrary, only by score, without having a reference to compare!! The X-ray ligand should have displayed the reference to compare scores and finally select compounds!
We thank to the reviewer for this comment. Based on the best docking score (more towards negative energy means high binding affinity) selecting ligand molecules is well known method from a decade, in SARS-CoV-2 in silico studies also have lots of example where scientists select best ligand molecules based on “best docking score” or “top hit”, here we follow the gold standard procedure in terms of the in-silico study derived by the previous researchers[1-4], the molecules were not chosen arbitrarily. We also repeated our docking study 8 times and got the similar values. Also, on comparison with reference is not possible because there is no such study. According to the reviewer’s suggestion X-ray ligands were used for the reference, where we found that the 11a and 11b poses -7.2 kcal/mol, and -7.5 kcal/mol which is much lower than our docked and selected top hit ligands. This section is now updated in the main manuscript and the supplementary as well according to the reviewer’s suggestion.
- Section 2.2. Molecular docking – the authors present the same information both in a detailed discussion of section 2.2 and Table 1 without indicating significant information on the investigated compounds. I think it was very useful if the authors briefly presented the amino acids known to be significant for SARS-CoV-2 inhibitory activity and then compare the interactions made by their compounds (organic triazole) with the known interaction ones. Moreover, the authors should have compared the interactions made by those four compounds with that of the x-ray ligand (6MOK).
This part is updated, and the glimpse of compounds are now discussed in the respective section. The interacting amino acids of Mpro with our ligand molecules were analyzed. In the crystal-ligand molecules 11a and 11b it is clearly visible that the 17 amino acids participating or present in Mpro inhibition pocket in the presence of 11a and 11b, on the other hand 7 Mpro amino acids are involved in the interaction with our four best ligand molecules chosen by the docking method. According to the reviewers’ suggestions the amino acid comparison with the crystal ligand - Mpro part is now included in the main manuscript. The ligands from crystal 11a and 11b gave only -7.2 kcal/mol and -7.5 kcal/mol binding energy which is much lower than our ligands, which clearly showed that our selected triazole based ligands can be more promising in use of Mpro inhibition. This data is now added in supplementary file.
- I don't understand exactly, why did the authors predict the binding pocket volume and surface area if they already have a ligand with the proper binding site?
We did the prediction to understand the binding pocket and the participating amino acids what was not clear in the previous study. Moreover, the previous study did not suggest appropriate pocket or participating amino acid numbers or the volume of the pocket, so getting more information about the binding pocket area with participating amino acid numbers and volume what we were going to use to prepare the docking box (important to select the region in protein where we wanted to perform the docking or the binding the ligands) the pocket analysis was performed. The detailed list of amino acids participating in pocket formation is in the supplementary file what gave much more clear view for further steps (when we finish our study no one gave detailed study results about the pocket, volume, etc. And in the end one most important thing is that the SARS-CoV-2 proteins are not well studied yet so getting up to mark information about the proteins, their inhibitory site, or pocket is another challenge.
- Moreover, a final conclusion of section 2.2 is missing. The authors, as I mentioned, only list some remarks.
This section is updated now in the main manuscript with proper conclusion as follows,
“In our study the ligands 11a and 11b (crystalized ligand structure used as inhibitor of Mpro in previous study) [25] were also docked against Mpro for assessment purpose. The 11a and 11b inhibitory ligands docking scores is low (-7.2 kcal/mol and -7.5 kcal/mol, Table S5), whereas our best triazole ligands showed binding affinities of -10.2 kcal/mol (Bemcentinib (DB12411)), -9 kcal/mol (Bisoctrizole:DB11262), -8.8 kcal/mol (PYIITM:DB07213), -8.8 kcal/mol (NIPFC:DB07020). Previous study suggests that the 17 (thr25, thr26, his41, cys44, met49, phe140, asn142, gly143, cys145, his163, his164, met165, glu166, pro168, asp187, arg188, gln189) amino acids were participating or present in the Mpro and inhibitory ligands interaction. Our protein-ligand interaction study suggested that the 7 amino acids (thr25, thr26, his41, cys44, met49, asn142, gln189) involved in Mpro inhibition are also involved in Mpro – Bemcentinib, Mpro – Bisoctrizole, Mpro – PYIITM, and Mpro – NIPFC interaction, which indicates that all four triazole based ligands have binding affinity with amino acids which plays crucial role in Mpro inhibition. In this term it can be concluded that Bemcentinib, Bisoctrizole, PYIITM, and NIPFC can be used as potential Mpro inhibitor”.
- I strongly suggest to rewrite this section by taken to account the above comments.
This section is updated now in the main manuscript according to the reviewers’ suggestions.
- Figures 3 and 4 are not relevant. The chemical structures of the selected compounds could be integrated into Table 1 at the appropriate names otherwise it just takes up space in the manuscript. Please change accordingly.
The figure 3 is now removed in supplementary and figure 4 is also rearranged with figure 5.
- Figure 5 indicates clearly and visibly the interaction types. However, if the author's considered Figure 4 important, then mix Figure 4 and 5 into one. In this way, both the interactions and the binding orientation of the compounds in the active site are preserved.
Figure 4 and Figure 5 is now merged and arranged more clearly. Please see the updated text with the newly inserted figure, which is also submitted as individual file.
- The authors must be very careful about what they write. They show at the beginning of section 2.5 that, based on docking and pharmacokinetic analyzes selected four potential inhibitors of the SARS-CoV-2 protease! THIS IS COMPLETELY UNTRUE! The authors selected these four compounds after docking and then performed pharmacokinetic and MD analyses on them. So, these compounds have been selected before!!!! If not, the authors are invited to evaluate all 27 compounds by docking and pharmacokinetics and then decide the best compounds with possible antiviral activity and finally, to run MD. Correct accordingly!!
We thank to the reviewer for this comment. We apologies for our linguistic mistake. It’s true that the compounds were selected based on the best docking score, we tried to express the same, but for the linguistic mistake the confusion raised. This section is now corrected and rephrased in the main manuscript. Yes, the four compounds were selected after docking based on their best docking score and proceed for the further analysis.
- MD Simulation - the authors discussed the H-bonds observed for Bemcentinib, Bisoctriazole, and PYIITM complexes during the 40s, but nothing about the NIPFC complex (just a picture)? For the latter, no interaction was observed during the 40ns?
It was a mechanical mistake from our side during the manuscript preparation in MDPI Molecules template. The MD studies suggest two hydrogen bonds consistency for NIPFC which also support our docking analysis study. This part is now corrected in the main manuscript.
- For MD simulation, the authors presented a great number of pictures which loads the article a lot? The same observation for the number of tables in section 2.3. I strongly recommend for the MD section that information such as RMDS or RMSF, or Rg, etc of all four compounds be plotted together. For example, when all 4 RMSFs are overlapped in a single graph, it may be easier to analyze. The same for RMSD or Rg, etc.
All four compounds MD Simulation data is rearranged and plotted according to the reviewer’s valuable suggestion and incorporated in the main manuscript. Please see the corrected part.
- The authors mentioned in Conclusions only 3 out of 4 compounds as potential SARs-COV-2 candidates. Why the authors excluded NIPFC compounds? On the contrary, if it is a compound that should be predicted with caution, it is Bemcentinib (see MD evaluation – some of the H-bond interaction have disappeared compared to the docking)
We apologies for our technical error. Now all four compounds are now in conclusion. In docking also Bemcentinib showed one hydrogen bond and in the end of the simulation also one hydrogen bond present (in plot the black one). Please see the conclusions sentences update:
The pharmacological profiling, docking analysis, MD simulation, MD trajectory, and interaction energy studies indicated that, Bemcentinib, Bisoctriazole, PYIITM and NIPFC could be used as possible drug candidates for inhibition against the SARS-CoV-2 Mpro protein to interrupt the essential role it plays in processing polyproteins translated from viral RNA. Based on the data presented in this paper, the compounds investigated in this study could be considered for further clinical studies and thereafter for potential treatment of COVID-19.
- Saha, S.; Nandi, R.; Vishwakarma, P.; Prakash, A.; Kumar, D. Discovering Potential RNA Dependent RNA Polymerase Inhibitors as Prospective Drugs Against COVID-19: An in silico Approach. Frontiers in Pharmacology 2021, 12, doi:10.3389/fphar.2021.634047.
- Eweas, A.F.; Alhossary, A.A.; Abdel-Moneim, A.S. Molecular Docking Reveals Ivermectin and Remdesivir as Potential Repurposed Drugs Against SARS-CoV-2. Frontiers in Microbiology 2021, 11, doi:10.3389/fmicb.2020.592908.
- Hosseini, M.; Chen, W.; Xiao, D.; Wang, C. Computational molecular docking and virtual screening revealed promising SARS-CoV-2 drugs. Precision Clinical Medicine 2021, 4, 1-16, doi:10.1093/pcmedi/pbab001.
- Choudhary, S.; Malik, Y.S.; Tomar, S. Identification of SARS-CoV-2 Cell Entry Inhibitors by Drug Repurposing Using in silico Structure-Based Virtual Screening Approach. Frontiers in Immunology 2020, 11, doi:10.3389/fimmu.2020.01664.

Reviewer 3 Report
A sound study which can be published without significant changes.
Some minor editing, mainly of language, is recommended. Things that caught my eye include (based on line numbers):
30 - word spacing
73 - colon unnecessary
71-80 - the section could be better organized / reordered
93 - too accurate (too many digits after decimal point)
118-119 - unnecessary upper case
154 - some text in the figure too small to read
159-160 - plural/singular mismatch
190 - comma
360 - ordering of value (tolerance) and unit
440 - comma
Author Response
A sound study which can be published without significant changes.
Some minor editing, mainly of language, is recommended. Things that caught my eye include (based on line numbers):
30 - word spacing
73 - colon unnecessary
71-80 - the section could be better organized / reordered
93 - too accurate (too many digits after decimal point)
118-119 - unnecessary upper case
154 - some text in the figure too small to read
159-160 - plural/singular mismatch
190 - comma
360 - ordering of value (tolerance) and unit
440 - comma
We thank to the reviewer for reviewing our manuscript. All raised comments are incorporated in point by point manner, for your kind consideration. Additionally, we had checked the corrected the English language of our manuscript. We had also highlighted the changes using the “Track of Changes” function, as per journal requirement.
- 30 - word spacing
We had corrected the word spacing in line 30 and also throughout the manuscript.
- 73 - colon unnecessary
The unnecessary colon was removed.
- 71-80 - the section could be better organized / reordered
The section has been reconstructed according to your valuable suggestions.
The new section is as follow:
“Till date, no specialized drugs are available on the market to cure COVID-19. Over recent years, the triazole group-based ligands have attracted the interest of the scientific community due to their comprehensive and multipurpose medicinal applications. Re-ports have been published stating that this group of ligands have potential antiviral, antibacterial, antifungal, antiparasitic and anti-inflammatory applications. Moreover, owing to the nature of their chemical properties, this group of ligands can be easily synthesized. The triazole group-based ligands could be a potential drug-candidate for use against the SARS-CoV-2 virus. Efforts to develop efficient therapeutic strategies against COVID-19 are still in progress.
In this study, we had evaluated the potential of the triazole ligands as effective antiviral agents. We identified the most suitable anti-SARS-CoV-2 candidate chemicals (based on their molecular docking scores), which were then further analyzed for posi-tive ADMET properties. Scientists across the world are researching different antiviral compounds, to identify those with the highest potential effectivity against SARS-CoV-2 as well as having low or no toxicity for humans. Our results suggest that the recom-mended drugs in this study, may be candidates for use in the treatment of COVID-19. Even though triazole ligands are already clinically approved drugs, they would still require clinical trials prior to repurposing as anti-COVID-19 medicines.”
- 93 - too accurate (too many digits after decimal point)
The number of the digits after the decimal point has been changed. The new sentence is as follow:
“Binding pocket volume of Mpro was 402.7 (SA) (Figure 2), respectively which signifies an optimum space for ligand binding.”
- 118-119 - unnecessary upper case
The name of the compound has been changed to:
“(5-{3-[5-(Piperidin-1-Ylmethyl)-1h-Indol-2-Yl]-1h-Indazol-6-Yl}-2h-1,2,3-Triazol-4-Yl)methanol”
- 154 - some text in the figure too small to read
The figure quality is improved and enlarged the writings within the figure too. And according to the another reviewer’s instruction figure 4 and figure 5 is merged now. Please see the updated text and the figure.
- 159-160 - plural/singular mismatch
The sentence is now corrected as per reviewer’s suggestion.
- 190 – comma
The sentence has been modified, by removing the comma.

Round 2
Reviewer 1 Report
Accept
Author Response
We thank to the reviewer for taking time to reviewing our manuscript carefully and accepting our manuscript. The comments provided by the respected reviewer has improved the quality of the manuscript. We had also checked the spellings and the English language of our manuscript.
Reviewer 2 Report
In attention of the journal Authors,
The authors responded satisfactorily to all referee’s requirements and made all the changes addressed in the manuscript, but still minor changes are needed. The manuscript has been substantially improved in both chemical content and English. The computational workflow, the software’s together with the results analysis, interpretation, and presentation employed to achieve their goal recommend this manuscript (“molecules-1396478 – v2”) as presenting scientific soundness and interest for the readers focused on triazoles and their use as possible agents to combat SARS-CoV-2.
In this context, I agree that the manuscript should be accepted for publication in Molecules journal, after a minor revision which is related only to the text consistency and the assembly of several figures into one and not to the chemical content.
Recommendations for the authors:
- Add a comma after in our study from the sentence: “In our study, the ligands 11a and 11b (crystalized ligand structure………..”
- For text consistency, capitalize each first letter of the aminoacids from the following sentences: “Previous study suggests that …..(thr25, thr26, his41, cys44, met49, phe140, asn142, gly143, cys145, his163, his164, met165, 174 glu166, pro168, asp187, arg188, gln189)……. Our protein-ligand interaction study (thr25, thr26, his41, cys44, met49, asn142, gln189)…………….”
- Please rewrite in a clear version (split the sentence in two), the following sentence: “ Our protein-ligand interaction study that the 7 amino acids (thr25, thr26, his41, cys44, met49, asn142, gln189)present involved in Mpro inhibition......................................”
- The following sentences could be mixed into one in order to not repeat: “Four best ligand molecules were selected based on the top hit for further analysis. Four organic triazole-based compounds were further analyzed for molecular interactions with SARS-CoV-2 (Mpro) (Table 1, Figure S13).”
Suggestion: “Four best ligand molecules were selected based on the top hit criteria and were further analyzed for molecular interactions with SARS-CoV-2 (Mpro) (Table 1, Figure S13).”
- MD simulations section, please combine all figures related to RMSD, RMSF, and Rg into one, as well as for H-bonds, SASA, and Interaction energy. By combining them, the reader will have a better perspective on the results analysis and implicitly will reduce the number of figures and will improve the article structure. Change the figure numbers and captions accordingly.
- After combining the figures, It is recommended to add a few subsections in the MD simulation section, as follows:
e.g. 2.5.1 RMSD analysis, 2.5.2 The radius of gyration analysis, 2.5.3 RMSF analysis, 2.5.4 hydrogen bond occupancy, 2.5.5 Hydrophobic interactions (SASA) analysis
or
before each paragraph dedicated to one of these aspects, add in italics RMSD analysis, RMSF analysis and so on.
Author Response
We thank to the reviewer for carefully reviewing our manuscript. We had responded to the raised questions in point-by-point manner, for your kind consideration. Additionally, we had checked the corrected the English language of our manuscript. We had also highlighted the changes using the “Track of Changes” function, as per journal requirement.
Recommendations for the authors:
- Add a comma after in our study from the sentence: “In our study, the ligands 11a and 11b (crystalized ligand structure………..”
This part is corrected now according to the reviewer’s suggestion.
- For text consistency, capitalize each first letter of the aminoacids from the following sentences: “Previous study suggests that …..(thr25, thr26, his41, cys44, met49, phe140, asn142, gly143, cys145, his163, his164, met165, 174 glu166, pro168, asp187, arg188, gln189)……. Our protein-ligand interaction study (thr25, thr26, his41, cys44, met49, asn142, gln189)…………….”
We thanks to the reviewer for this comment. This part is corrected now according to the reviewer’s suggestion.
- Please rewrite in a clear version (split the sentence in two), the following sentence: “ Our protein-ligand interaction study that the 7 amino acids (thr25, thr26, his41, cys44, met49, asn142, gln189)present involved in Mpro inhibition......................................”
Thank you so much for your suggestion. This part is corrected now according to your suggestion, for the better understanding of the readers.
- The following sentences could be mixed into one in order to not repeat: “Four best ligand molecules were selected based on the top hit for further analysis. Four organic triazole-based compounds were further analyzed for molecular interactions with SARS-CoV-2 (Mpro) (Table 1, Figure S13).”
We thanks to reviewer for this suggestion. We agree with the reviewer’s suggestion. This part has been modified accordingly.
Suggestion: “Four best ligand molecules were selected based on the top hit criteria and were further analyzed for molecular interactions with SARS-CoV-2 (Mpro) (Table 1, Figure S13).”
- MD simulations section, please combine all figures related to RMSD, RMSF, and Rg into one, as well as for H-bonds, SASA, and Interaction energy. By combining them, the reader will have a better perspective on the results analysis and implicitly will reduce the number of figures and will improve the article structure. Change the figure numbers and captions accordingly.
We thanks to the reviewer for this comment. All MD figures are now merged and represented in one panel and the figure numbers are changed accordingly.
- After combining the figures, It is recommended to add a few subsections in the MD simulation section, as follows:
e.g. 2.5.1 RMSD analysis, 2.5.2 The radius of gyration analysis, 2.5.3 RMSF analysis, 2.5.4 hydrogen bond occupancy, 2.5.5 Hydrophobic interactions (SASA) analysis
or
before each paragraph dedicated to one of these aspects, add in italics RMSD analysis, RMSF analysis and so on.
The section is updated now with a paragraph wise heading as per reviewers’ suggestion.